# Two novel and correlated CF-causing insertions in the (TG)mTn tract of the CFTR gene

Silvia Pierandrei[1], Giovanna Blaconà[2], Benedetta Fabrizzi[3], Giuseppe Cimino[4], Natalia Cirilli[3], Nicole Caporelli[3], Antonio Angeloni[2], Marco Cipolli[3], Marco Lucarelli[2,5]*

**1** Dept. of Mother-Child and Urologic Sciences, Sapienza University of Rome, Rome, Italy, **2** Dept. of Experimental Medicine, Sapienza University of Rome, Rome, Italy, **3** Cystic Fibrosis Care Center, Mother - Child Department, United Hospitals, Ancona, Italy, **4** Cystic Fibrosis Care Center, Umberto I Hospital, Rome, Italy, **5** Pasteur Institute Cenci Bolognetti Foundation, Rome, Italy

* marco.lucarelli@uniroma1.it

**Data Availability Statement:** All relevant data are within the manuscript and Supporting Information files.

## Abstract

Two novel and related pathogenic variants of the Cystic Fibrosis Transmembrane conductance Regulator (CFTR) gene were structurally and functionally characterized. These alterations have not been previously described in literature. Two patients with diagnosis of Cystic Fibrosis (CF) based on the presence of one mutated allele, p.Phe508del, pathological sweat test and clinical symptoms were studied. To complete the genotypes of both patients, an extensive genetic and functional analysis of the CFTR gene was performed. Extensive genetic characterization confirmed the presence of p.Phe508del pathogenic variant and revealed, in both patients, the presence of an insertion of part of intron 10 in intron 9 of the CFTR gene, within the (TG)m repeat, with a variable poly-T stretch. The molecular lesions resulted to be very similar in both patients, with only a difference in the number of T in the poly-T stretch. The functional characterization at RNA level revealed a complete anomalous splicing, without exon 10, from the allele with the insertion of both patients. Consequently, the alleles with the insertions are expected not to contribute to the formation of a functional CFTR protein. Molecular and functional features of these alterations are compatible with the definition of novel CF-causing variants of the CFTR gene. This also allowed the completion of the genetic characterization of both patients.

## Introduction

Cystic Fibrosis (CF) is an autosomal recessive disease caused by pathogenic variants of Cystic Fibrosis Transmembrane conductance Regulator (CFTR) gene [1–4]. A high intragenic variability, due to a huge number of CFTR variants combined in *trans*, is often enhanced by the presence of two or more variants in *cis* to constitute haplotypes and/or complex alleles [5, 6]. Additional genetic variability is also introduced by modifier genes, which can affect the severity of the CF phenotype through, for example, an alternative chloride conduction, the regulation of splicing, the modulation of CFTR gene expression and the modulation of susceptibility

**Funding:** This work was supported by Regione Lazio (research projects 2008 – 2012). The funders had no role in study design, data collection and analysis, decision to publish, or preparation of the manuscript.

**Competing interests:** The authors have declared that no competing interests exist.

to infections/inflammatory responses [7, 8]. The intra- and extra-CFTR genetic variability, also combined with many other factors (*i.e.* environment, treatment approach, patient's adherence to therapy) originates a complex genotype–phenotype relationship in CF that, in turn, affects diagnosis, prognosis and personalized therapy [2, 3, 9–15].

From clinical point of view, CF has been reinterpreted as characterized by a wide spectrum of clinical manifestations spanning from mono-symptomatic forms often limited to male reproductive apparatus [16–18] to poly-symptomatic severe forms involving several organs, mainly lungs and pancreas [8]. This clinical variability has been organized within clinical categories, which distinguish between CF and CFTR-related disorders (CFTR-RD) [19]. The advent and the spread of newborn screening programs for CF since the late '70es started identifying a new diagnostic designation called cystic fibrosis screening positive inconclusive diagnosis (CFSPID) or cystic fibrosis related metabolic syndrome (CRMS) [20]. Sometimes, these forms can evolve in CFTR-RD or even classical CF pictures [21] and for this reason they need follow-up arrangements.

Diagnostic criteria for CF are well defined in guidelines and consensus documents [22–24] and are based on at least two sweat chloride value $\geq$ 60mEq/L and/or two CF-causing variants, pancreatic insufficiency or other suggestive symptoms. CFTR-RD diagnosis can be posed in presence of persistent intermediate sweat chloride values, inconclusive genetic analysis and single organ involvement [19] that is acute recurrent/chronic pancreatitis or recurrent pneumonia ($\geq$ 2 episodes in one year; $\geq$ 3 episodes throughout life) or disseminated bronchiectasis or nasal polyposis or sinusitis or male infertility for congenital bilateral absence of vas deferens (CBAVD). The CFSPID diagnosis can be posed in presence of asymptomatic infants, positive to neonatal screening, with sweat chloride values persistently in the borderline range with one or no pathogenic CFTR gene variants, or with sweat chloride values in the normal range with two pathogenic CFTR gene variants, at least one of which with unclear phenotypic consequences [20].

A crucial point of CF genetics is the still unclear functional and clinical meaning of most of CFTR variants. A great international effort is underway to functionally and clinically characterize the greatest possible number of variants [6, 14, 25, 26]. Genetic information about the CFTR gene and its variants can be found in two open access databases. The CFTR1 database [http://genet.sickkids.on.ca] provides, since 1989 up to date, structural information about individual variants of CFTR gene, with more than 2000 variants listed. In 2010 this project was upgraded and the CFTR2 database [https://cftr2.org/] was created with the specific aim to provide a comprehensive reviewed functional and clinical information on CFTR variants. At moment, a total of 412 CFTR variants are functionally annotated in the CFTR2 database.

Moreover in the era of new personalized CFTR modifying therapies, CFTR gene analysis is an important inclusion criteria for the selection of patients in clinical trials or in treatment protocols [27]. Due to the importance of this topic an ECFS initiative is ongoing [https://www.ecfs.eu/content/complete-cftr-gene-mutation-analysis-european-patients-cystic-fibrosis]. Despite of the increasingly powerful methods of mutational search [28–30], some patients remain without both pathogenic variants found even after in-depth genetic analysis, with diagnostic, prognostic and therapeutic impairment [31].

Here we present two clinical cases diagnosed as CF based on positive sweat tests and severe clinical symptoms, with one p.Phe508del mutated allele evidenced at a first level genetic test. These patients share similar clinical features: early-onset exocrine pancreatic insufficiency (PEI), absence of radiological features of CF during early childhood and a first isolation of *P. Aeruginosa* during early childhood. A full genetic and functional analysis allowed to select and characterize two novel insertions, strictly related, in the (TG)mTn tract of the CFTR gene. Both insertions originate a full anomalous splicing, which produces a non-functional CFTR

mRNA without exon 10. These findings point to two novel CF-causing variants of the CFTR gene.

## Materials and methods

### Biochemical, microbiological and clinical characterization of patients

The biochemical, microbiological and clinical features of patients are summarized in Table 1.

The first case, a female born in 2001 (patient 1), was diagnosed for symptoms at the age of 10 months; at that time newborn screening program for CF was not yet introduced in her region of origin. Family history of CF was negative. She developed gastrointestinal symptoms (underweight and diarrhoea) shortly after birth. The symptoms did not improve neither weaning from breast milk to a soy formula because of a suspected cow's milk protein allergy, nor weaning from soy formula to rice formula because of a suspected soy allergy. Serologic testing for coeliac disease, faecal occult blood test and stool and urine cultures were negative. After an episode of pseudo-Bartter's syndrome (loss of salts) at 10 months CF was strongly considered. Sweat chloride values of 80 and 92 mmol/L confirmed the diagnosis. A first level genetic analysis showed the presence of only the c.1521_1523delCTT, p.Phe508del (legacy name F508del) pathogenic variant in heterozygosis. The segregation analysis demonstrated the paternal origin of the p.Phe508del pathogenic variant (absent in the mother). Faecal chymotrypsin activity was undetectable and she developed PEI. Consequently, a pancreatic enzyme replacement therapy was initiziated improving her nutritional status. To date, the patient body mass index

**Table 1. Biochemical, microbiological and clinical features of patients.**

| | Patient 1 | Patient 2 |
|---|---|---|
| **General and biochemical features** | | |
| Gender | F | M |
| Current age | 18 years | 7 years |
| Age at diagnosis | 10 months | Birth |
| Newborn screening (IRT dosage) | Non-available | 69 ng/mL (positive) |
| Sweat chloride | 80 and 92 mmol/L | 95 and 102 mmol/L |
| Diagnostic criteria for CF | Clinical features (pancreatic exocrine insufficiency (PEI) and salt loss syndrome) and sweat chloride value ≥60 mmol/L. | Positive NBS, clinical features (PEI) and sweat chloride value ≥60 mmol/L. |
| **Gastrointestinal features** | | |
| Pancreatic exocrine insufficiency (PEI) | Yes (birth) | Yes (birth) |
| Current pct BMI (CDC) | 20° | 43° |
| Distal intestinal obstruction syndrome (DIOS) | No | 1 episode |
| CF-associated liver disease | Yes (8 years) | No |
| **Respiratory complications** | | |
| CF-associated pathogens | Chronic infection with *S. aureus* (methicillin-sensitive, MSSA) Intermittent infection with *P. aeruginosa* | Chronic infection with *S. Aureus* (methicillin-sensitive, MSSA) Intermittent infection with *P. aeruginosa* |
| First isolation of *P. aeruginosa* | 8 years | 3 years |
| Current computed tomography (CT) scanning | Upper lobes, middle right lobe and lingula bronchiectasis and air trapping; nor mucus plugging, nor airway wall thickening nor atelectasis, nor consolidation | Neither bronchiectasis, nor mucus plugging, nor airway wall thickening nor atelectasis, nor consolidation |
| Current FEV1 | 115% | 89% |
| Allergen sensitization profile | No | Mites and Alternaria |

(BMI) percentile is 20˚ according to Centers for Disease Control and Prevention (CDC) growth chart. Throat swabs cultures showed chronically colonisation with methicillin-sensitive *S. aureus* (MSSA) from 2015 and intermittent infection with P. aeruginosa (first isolation in 2009). Bronchiectasis affecting upper lobes, right middle lobe and lingula has been observed on chest computed tomography (CT) for the first time on October 2018 (not identified during the previous one on October 2016). Her last forced expiratory volume in 1 second (FEV1) resulted to be 115% of predicted (3.7 liters).

The second case was a male born in 2012 (patient 2). The child was adopted soon after birth. Information about patient's biological family health history and ancestry are lacked, but he shares the same geographical origin of the patient 1 (south of Marche Region, Central Italy). CF diagnosis was made by positive newborn screening (IRT, 69 ng/mL), positive sweat test (95 and 102 mmol/L) and PEI (pathological levels of faecal elastase 1). A first level genetic analysis detected only the c.1521_1523delCTT, p.Phe508del (legacy name F508del) pathogenic variant in heterozygosis. Early pancreatic enzyme replacement therapy ensured an adequate nutritional status and his current BMI (CDC growth chart) is 43˚. He developed one episode of distal intestinal obstruction syndrome (DIOS) in 2016. Elevated total IgE (range: 2000–5000 UI/ml) has been found since 2014. Further analysis has only identified an atopic state with *Alternaria* and *Mites* sensitization. Throat swabs cultures showed chronically colonisation by *S. aureus* (MSSA) and intermittent infection by *P. aeruginosa* (first isolation in 2015). No CF imaging alteration are detected by CT. His last FEV1 resulted 89% of predicted (1.27 liters).

To complete the genotypes of both patients, extensive genetic and functional analyses of CFTR gene were performed (see below).

## Sweat test and pancreatic status evaluation

Both patients underwent a sweat test twice, performed by a quantitative pilocarpine iontophoresis method [32], by using the Macroduct device (Delcon, Milan, Italy) for sweat collection and the Jenway PCLM3 analyser (VWR International, Milan, Italy) for chloride measurement. In accordance with recent guidelines [22], the sweat test was considered positive for both patients because all values were > 60 mEq/L.

For patient 1, exocrine pancreatic function was evaluated by the dosage of stool chimotrypsin [33] (test Chymo, Roche, Mannheim). A pathological level of stool chimotrypin (< 4.2 U/g) was determined in at least two independent dosages. For patient 2, exocrine pancreatic function was evaluated by the dosage of stool elastase 1 [34] (test PANEL FE Elastase Elisa, Bioserv Diagnostics, Germany). A pathological level of fecal elastase 1 (< 200 μg/g) was determined in at least two independent dosages.

## CFTR mutational analysis

Genomic DNA was extracted from peripheral blood leukocytes by the QIAamp DNA Blood midi kit (Qiagen, Hilden, Germany) and quantified using a fluorimeter (Qubit, Invitrogen, CA, USA). The first level genetic analysis of CFTR gene (RefSeq NM_000492.3, NG_016465.3) was performed by INNO-LIPA CFTR19, CFTR17+Tn Update and CFTR Italian Regional kit (Fujirebio Europe). A second level genetic analysis was performed by sequencing. The proximal 5'-flanking, all exons and adjacent intronic zones were PCR-amplified and sequenced by a Sanger cycle sequencing protocol (ThermoFisher scientific, Waltham, MA, USA) in a 96-well format already published by us [35], using a genetic analyzer (ABI PRISM 3130*xl*; Applied Biosystems, Foster City, CA, USA). For data analysis, a specific template for SeqScape software version 2.7 (Applied Biosystems) were used [36]. A third level genetic analysis was performed by multiplex ligation-dependent probe amplification (MLPA) (MRC-Holland). The primers

**Table 2. Primers used for genomic DNA amplification and sequencing.** Pairs F1-R1 and F2-R2 were used for PCR amplification. Primers F1, F2 and F3 were used for forward sequencing; primers R1, R2 and R3 were used for reverse sequencing. The position of the nucleotide at 5'-end of each primer is indicated in both legacy and HGVS name. The annealing temperatures used (Ta) and the length of amplicons are indicated.

| Name | 5'-end position (Legacy name) | 5'-end position (HGVS name) | Forward primer | Name | 5'-end position (Legacy name) | 5'-end position (HGVS name) | Reverse primer | Ta | Length |
|---|---|---|---|---|---|---|---|---|---|
| F1 | 1342–219 | c.1210-219 | 5'-TTGATAATGGGCAAATATCTTAG-3' | R1 | 1524+81 | c.1392+81 | 5'-CCTTCCAGCACTACAAACTA-3' | 54˚C | 483 bp |
| F2 | 1525–2794 | c.1393-2794 | 5'-CCACTACGCCCGGCTAATT-3' | R2 | 1525–2549 | c.1393-2549 | 5'-AATTCATGGAAAGCTTGTTTGG-3' | 58˚C | 245 bp |
| F3 | 1342–34 | c.1210-34 | 5'-TGTGTGTGTGTGTTTTTTTTTTTTTTTTTTTTT-3' | R3 | 1525–2903 | c.1393-2903 | 5'-GCGACAGAGCGAGACTCCGTCTCAAA-3' | 60˚C | - |

specifically used for the analysis of the insertions are reported in Table 2. In particular, the pair F1-R1 was used for PCR amplification of the zone of the insertion. To analyze both the wild type and the anomalous amplicons evidenced at the PCR step, they were separately extracted and purified from 1% agarose gel by the QIAquick Gel Extraction kit (Qiagen) and individually sequenced by a Sanger cycle sequencing protocol (ThermoFisher). In particular, primers F1 and F3 were used for forward sequencing and primers R1 and R3 for reverse sequencing, of purified amplicons. The hypothesis that the molecular lesion was duplicated (and not transferred from intron 10 to intron 9, respectively intron 9 and intron 8 in previous nomenclature) was verified by sequencing the zone of interest in intron 10. This was achieved by using specific primers for PCR amplification (Table 2, pairs F2-R2) and then by extraction, purification and Sanger cycle sequencing of amplicons (Table 2, primer F2 for forward sequencing and primer R2 for reverse sequencing). The primer R2, external to the inserted zone, ensured the PCR amplification of the wild type zone of intron 10.

The primers used for exon 10 analysis are located in the preceding and following introns of exon 10 to span the entire exon 10 and previous and subsequent adjacent intronic zones. This large positioning of primers and the negativity of MLPA analysis excluded the possibility that results could be influenced by some duplication of part of exon 10 at multiple locations of probands' genome.

## Fragment analysis

To confirm the right size of the molecular alterations, a DNA fragment analysis protocol (ThermoFisher) was optimized. A specific oligonucleotide (forward: 5'-TTGATAATGG GCAAATATCTTAG-3') was labeled with fluorescent dye (dROX; Applied Biosystems) and used for PCR amplification (paired with the following reverse primer: 5'-CCTTCCAGCACT ACAAACTA-3') of the zone of interest. Labeled amplicons were separated by capillary electrophoresis on ABI PRISM 3130*xl* (Applied Biosystems) and analyzed using GeneMapper software version 4.1 (Applied Biosystems).

## CFTR expression analysis

Expression analysis was performed by a protocol that, using an optimized set of primers, allows the display of any possible anomalous splicing of CFTR mRNA, as previously described [37]. Briefly, starting from nasal brushing of patients, RNA was extracted by the RNeasy mini kit (Qiagen), reverse transcribed and amplified by RT-PCR (Bio-Rad, Hercules, CA, USA) producing 7 amplicons spanning the entire CFTR mRNA. All the 7 cDNA amplicons were extracted from agarose and individually sequenced as described above. The primers specifically used for the analysis of exon 10 splicing are reported in Table 3. In particular, one forward

**Table 3. Primers used for cDNA amplification and sequencing.** Pairs F4-R4 and F4-R5 were used for RT-PCR. Primer F4 was used for cDNA forward sequencing; primers R4 and R5 were used for cDNA reverse sequencing. The position of the nucleotide at 5'-end of each primer is indicated in both legacy and HGVS name. The annealing temperatures used (Ta) and the length of amplicons are indicated.

| Name | 5'-end position (Legacy name) | 5'-end position (HGVS name) | Forward primer | Name | 5'-end position (Legacy name) | 5'-end position (HGVS name) | Reverse primer | Ta | Length |
|------|------|------|------|------|------|------|------|------|------|
| F4 | 1257 | c.1125 | 5'–ACAAAAGCAAGAATATAAGACATTG-3' | R4 | 1800 | c.1668 | 5'–AATTCTTGCTCGTTGACCTCCACTC-3' | 60°C | 543 bp |
| F4 | 1257 | c.1125 | 5'–ACAAAAGCAAGAATATAAGACATTG-3' | R5 | 1602 | c.1470 | 5'–GAATGAAATTCTTCCACTGTGC-3' | 60°C | 346 bp |

primers (F4, located in exon 9) and two different reverse primers (R4, located in exon 12 and R5, located in exon 11) were used. All cDNA amplicons, spliced and unspliced, were recovered from agarose gel and forward sequenced by F4 primer and reverse sequenced by R4 or R5 primers. This allowed the evaluation of exon 10 splicing and of the presence of p.Phe508del pathogenic variant, as well as of their segregation on different alleles also at cDNA (RNA) level. Gel electrophoresis runs were scanned by a CCD camera (VisiDoc-It; UVP, Cambridge, UK) and analyzed on the VisionWorks LS software version 6.7.3 (UVP) for densitometric assays. Expression analysis results were confirmed by Real Time PCR.

## Statistical analysis

Expression data were evaluated using analysis of variance (ANOVA) by GraphPad Prism 5. A $p < 0.05$ was considered statistically significant.

## Ethics statement

The analyses were performed and the data were collected for diagnostic purposes. In particular, to complete the genotype of both patients and to confirm the diagnosis of Cystic Fibrosis. If the study is performed for diagnostic purposes it is not necessary, in our Institution, to seek the approval of the ethics committee.

We have, for both families, the informed consent to the genetic test and the authorization to publish the results.

## Results

In both patients, the genetic characterization performed by sequencing and MLPA confirmed the presence of p.Phe508del pathogenic variant in heterozygosis. No other known CF pathogenic variant was identified by these methods. The polymorphic (TG)mTn tract, known to modulate exon 10 splicing, appeared to be non-pathological. However, the amplification of the exon 10 and surrounding intronic regions of CFTR gene showed an extra amplicon greater (of about 300 nucleotides) than the WT in both patient 1 and patient 2 (Fig 1, lanes 1 and 3 respectively). This amplicon is absent both in the patient 1 father and in an individual of the general population (Fig 1, lanes 4, 5 respectively) while it is present in the patient 1 mother (Fig 1, lane 2). The dimensions of the molecular alterations of both patients resulted to be similar, with only a small increase in the length of that of patient 2 (Fig 1, lane 3).

The structure of anomalous amplicons was reconstructed by both fragment analysis and sequencing after their recovery from agarose gel. By fragment analysis, the migration profile of each WT sample was compared to the anomalous sample profiles. This allowed to determine the presence or absence of the molecular alterations and to have an estimation of their size independent from sequencing. We selected the most efficient primer which let to distinguish between the anomalous and the WT alleles in several tested samples. The size of all amplicons,

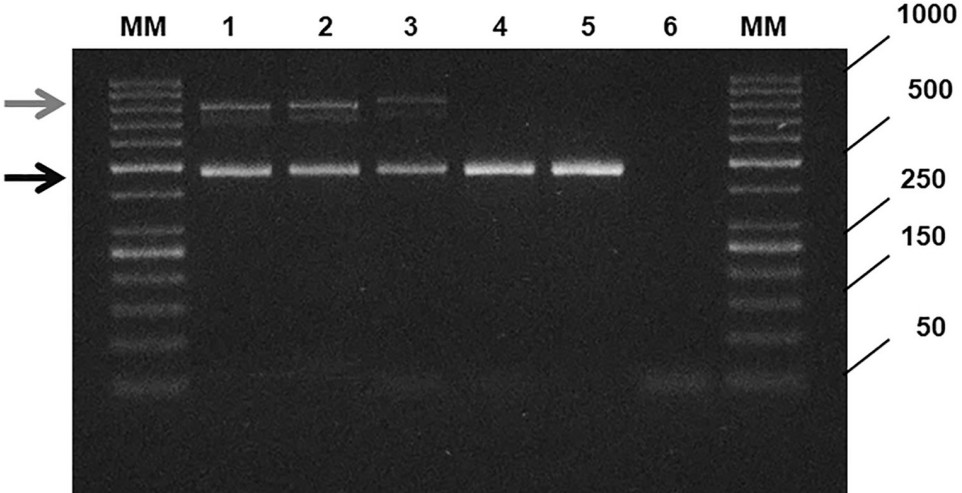

**Fig 1. DNA genomic amplification of CFTR exon 10 and adjacent intronic regions.** The WT amplicon (483bp) is indicated by the lower arrow. The anomalous amplicon (~800 bp) is indicated by the upper arrow. Lane 1: case 1; lane 2: case 1 mother; lane 3: case 2; lane 4: WT control (case 1 father); lane 5: WT control (general population); lane 6: PCR negative control; MM: DNA ladder. The extra bands visible in the first three samples (lane 1, 2 and 3) are due to the heteroduplex formation (as verified by a "reconditioning" procedure).

defined by DNA fragment analysis approach, resulted to be 483.9, 799.1 and 840.1 nucleotides respectively for WT, patient 1 and patient 2 amplicons (Fig 2, panels 1, 2 and 3 respectively). Each anomalous amplicon was sequenced by distal (Table 2; F1 for forward sequencing and R1 for reverse sequencing) and proximal (Table 2; F3 for forward sequencing and R3 for reverse sequencing) primers. In particular, the proximal primers (especially the F3 forward primer anchored to the end of (TG)m and to the start of Tn) allowed a better count of the number of T.

The data from sequencing and fragment analysis were elaborated by SeqScape software (Applied Biosystems) to reveal the exact structure of molecular lesions, as shown in Fig 3. The analysis showed a rearrangement at the level of the polymorphic (TG)mTn tract (Fig 2, panel 2 and 3), compared to the WT (Fig 2, panel 1). The presence of many repeated T, within the (TG)m repeat, followed by a sequence not corresponding to exon 10 of the CFTR gene was found. This repetition resulted longer in the patient 2 (76T, Fig 2, panel 3) than in the patient 1 (35T, Fig 2, panel 2). The different size between case 1 and case 2 anomalous amplicons resulted to depend only from the different number of T repeat. The alignment of DNA sequences with reference templates revealed, in both patients, the presence of an insertion of part of intron 10 within the (TG)m repeat of intron 9 of the CFTR gene, preceded by the poly-T stretch (Fig 2, panels 2 and 3). The portion of intron 10 inserted in intron 9 is the result of an anomalous DNA duplication, as verified using a specific PCR amplification targeted to the WT sequence of intron 10 (Table 2; primers F2 and R2). This allowed confirming the presence of WT intron 10 in both patients.

To determine if the molecular alterations independently segregate from the p.Phe508del pathogenic variant, parental DNA studies were made. However, they were possible only for patient 1. The patient 1 father carried the p.Phe508del pathogenic variant (but not the novel insertion) and the mother carried the novel insertion (but not the p.Phe508del). This confirmed that the genotype is compound heterozygous for the p.Phe508del pathogenic variant

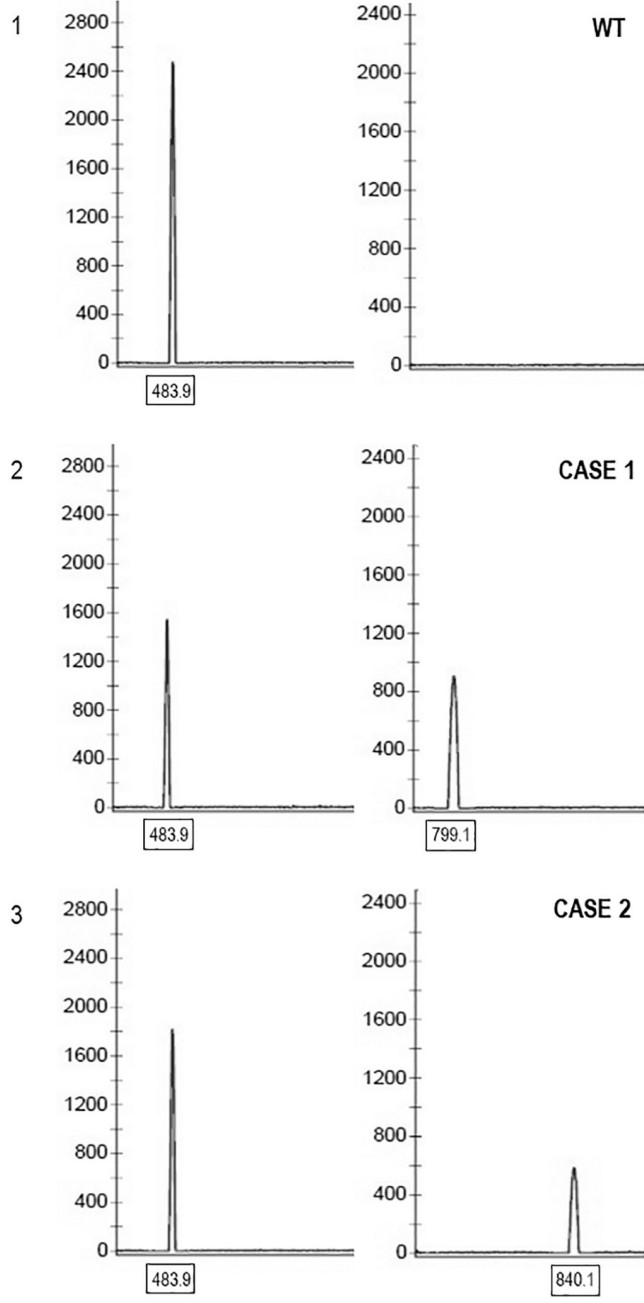

**Fig 2. Fragment analysis of CFTR Exon 10.** The panel 1 shows a WT peak of exon 10 (MW of obtained amplicon: 483.9) in a control sample; the panel 2 shows the case 1 peaks of exon 10: the WT peak (MW: 483.9) is represented on the left, whereas the peak with the molecular alteration characterized by an anomalous number of T (MW: 799.1) is shown on the right; the panel 3 shows the case 2 peaks of exon 10: the WT peak (MW: 483.9) is represented on the left, whereas the peak with the molecular alteration characterized by an anomalous number of T (MW: 840.1) greater than the case 1 is shown on the right.

on one allele and the novel insertion on the other. Although it was not possible to verify the allelic segregation in the case 2 parents, it is highly probable that also the case 2 is compound heterozygous for the other novel insertion. In addition, for both patients segregation analysis was also performed at cDNA level (see below).

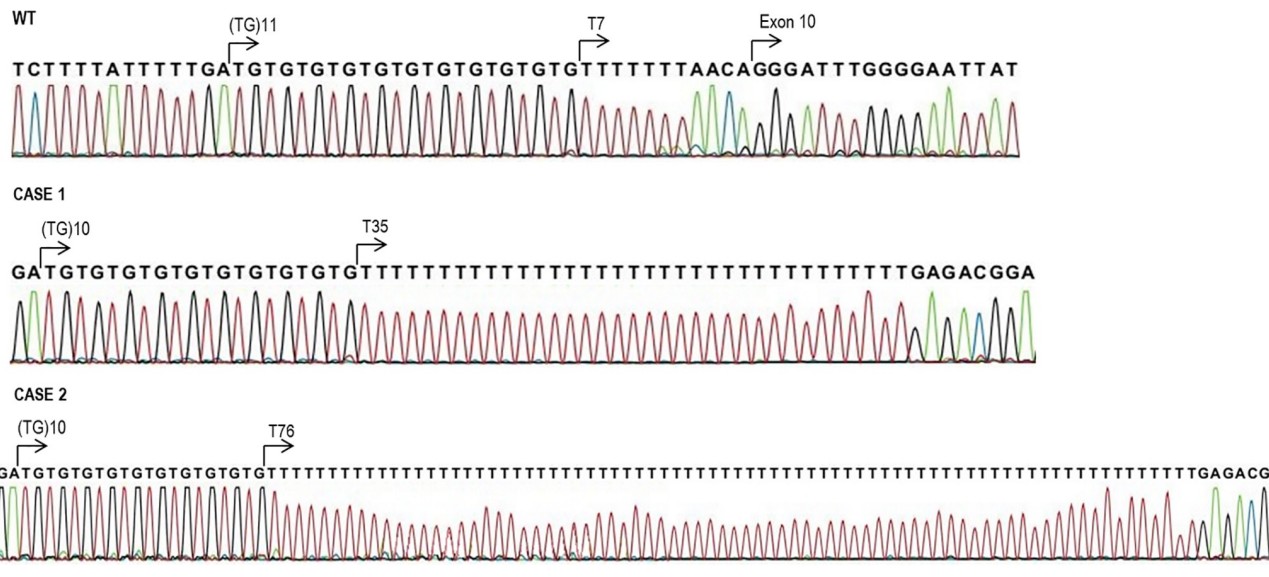

**Fig 3. (TG)mTn tract structure.** The panel 1 shows an example of a WT forward sequence of the (TG)mTn tract; the panel 2 shows the case 1 forward sequence characterized by an anomalous number of 35 T in respect to the WT; the panel 3 shows the case 2 forward sequence characterized by an anomalous number of 76 T in respect to the WT and greater than the case 1. For each sample, the exact number of T were defined by different combination of forward and reverse sequencing as well as by fragment analysis (shown in Fig 2). See text for explanation.

To verify the functional effect of the molecular lesions, the expression analysis of the entire CFTR mRNA was performed. All the amplicons of RNA analysis not including exon 10 resulted WT after recovery from agarose gel and sequencing. The amplicon including exon 10 revealed an alteration of mRNA structure. The functional characterization at RNA level, by RT-PCR and sequencing, revealed, in both cases, the presence of an anomalous splicing of exon 10 (Fig 4A, lanes 2 and 3, lower arrow). This aberrant splicing appeared to completely depend on the allele with the molecular alteration, as the other allele showed a non-pathological (TG)10T9 polymorphic tract in both patients. In fact, a homozygous (TG)10T9 polymorphic tract showed a complete absence of anomalous splicing in WT controls (Fig 4A, lane 1, upper arrow, showed as a representative example). In addition, for both patients, the sequencing of the lower and the higher cDNA bands shown in Fig 4 (after their separate recovery form agarose gel) evidenced the inclusion of exon 10 and the presence of p.Phe508del pathogenic variant in the higher band and the exclusion of exon 10 and the absence of p.Phe508del in the lower band. This confirmed for both patients the segregation of both novel insertions from the p.Phe508del pathogenic variant. A quantitative densitometric assay of both CFTR amplicons, with or without exon 10 were performed (Fig 4B). The percentage of total exon 10 anomalous splicing (Fig 4B, right panel) resulted to be 65.3% (±4.8) and 59.3% (±7.8) in the patient 1 and patient 2 respectively, compatible with the presence of one allele with the molecular alteration. Quantitative data of splicing were confirmed by real time PCR. Consequently, the mutated allele is expected not to contribute to the formation of functional CFTR protein. By contrast, in the WT control no exon 10 anomalous splicing was found.

These studies allowed to understand and schematize the structural organization of molecular alterations (Fig 5). It is an insertion that disrupt the canonical (TG)mTn splicing site, located at the level of intron 9 –exon 10 junction. In particular, a physiological repetition of (TG)10 is followed by an abnormal poly-T stretch, different in length in the two cases: 10 T in case 1 and 51 T in case 2. After this poly-T stretch a 306 bp portion of intron 10, starting with

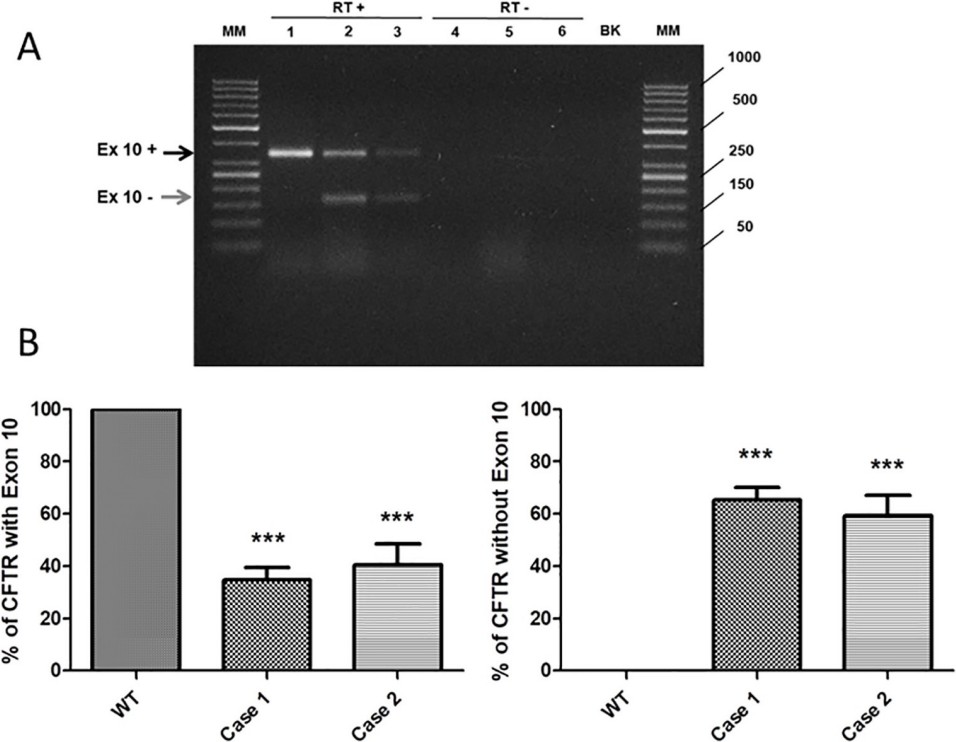

**Fig 4. Analysis of CFTR exon 10 splicing.** A) cDNA amplification of CFTR Exon 10. The amplification was performed with a pairs of primers spanning from exon 9 to exon 11 (see Materials and methods). The WT amplicon, including the exon 10 (346 bp), is indicated by the upper arrow (black). The anomalous amplicon, without the exon 10 (163 bp) is indicated by the lower arrow (grey). Lane 1: WT control (general population); lane 2: case 1; lane 3: case 2; lane 4, 5, 6: RT-PCR negative controls (with RNA, without retrotranscriptase); lane BK: RT-PCR negative control (without RNA); MM: DNA ladder. B) Quantification of both WT (left panel) and anomalous amplicon (right panel) by densitometric analysis. The proportion of the CFTR with Exon 10 results 100%, 34.7% and 40.7% for the control, the case 1 and the case 2, respectively. The proportion of CFTR without Exon 10 is 0%, 65.3% and 59.3% for the control, the case 1 and the case 2, respectively.

25 T, is inserted. Consequently, a total of 35 T and 76 T for case 1 and case 2 respectively was achieved. After the insertion, in both cases a G was found. The overall length of the molecular alterations were 317 bp for the case 1 and 358 bp for the case 2. At the end of the molecular alterations a (TG)4T7 repetition was found.

## Discussion

Currently, an extended CFTR mutational analysis in CF patients achieves up to 98% of detection rate (DR), leaving about 2% of genotypes with one or two unknown alleles [38]. This is an effect of the high genetic heterogeneity of the CFTR gene, which may be damaged by a lot of rare pathogenic variants. Due to their low frequency, these rare variants may escape even to mutational search protocol with high detection rate. In addition, several pathogenic variants have peculiar structural features and, consequently, may be not recognized during common mutational searches [39]. The very low frequency (even the individuality) and the structural peculiarity of the unknown alleles of the two cases described here are probably the reasons of their non-recognition at the initial genetic analysis. To date, although over 2000 variants of the CFTR gene are known, some escape the most common analysis protocols. In addition, only for a small part of these variants an experimental functional characterization has already been done.

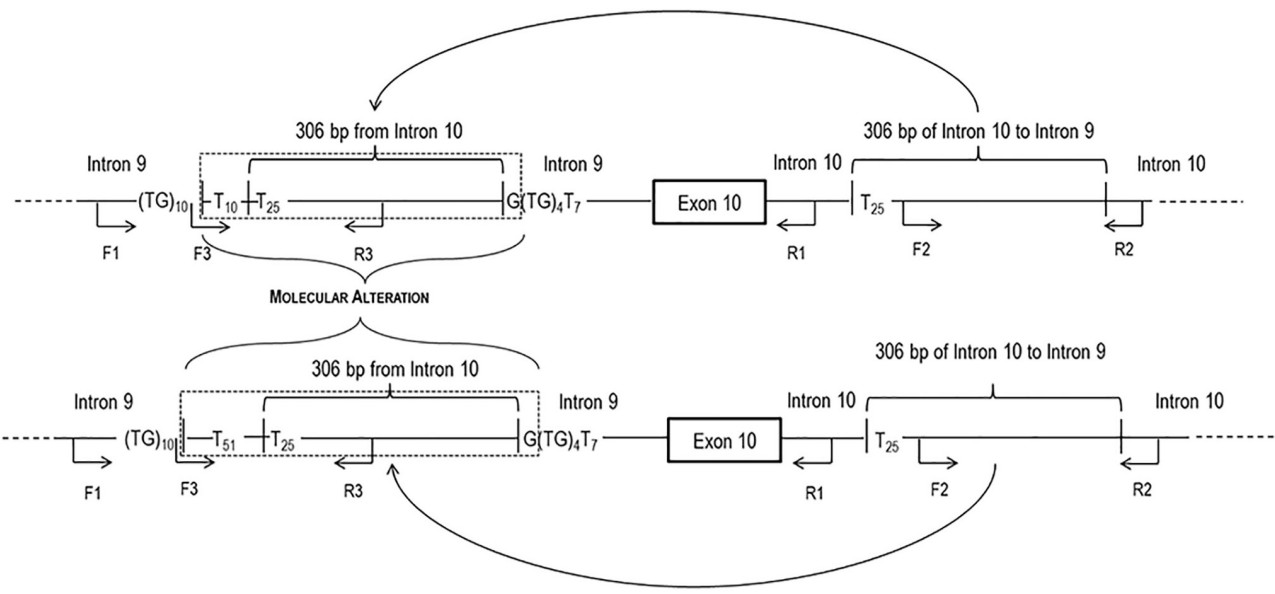

**Fig 5. Schematic representation of the molecular alterations.** At CFTR intron 9 level, within the (TG)mTn tract, a poly-T stretch followed by the insertion of 306 bp of CFTR intron 10 was evidenced. The molecular lesions resulted to be very similar in both patients, with only a small difference in the number of T in the poly-T stretch preceding the common part of the lesion (in particular 10 T in case 1 and 51 T in case 2). The figure also reports the position of primers (F1, F2, F3, R1, R2, R3) used and described in Table 2. See text for explanation.

The patients described here had a classical (TG)10T9 tract on one allele and a disrupted (TG)mTn tract on the other. As already known, the polymorphic (TG)mTn tract, located in the CFTR intron 9, regulates the splicing of CFTR exon 10. Variations of this tract may cause deleterious effects on the pre-mRNA splicing [40]. The two novel pathogenic variants correspond both to an insertion, within canonical (TG)mTn tract, of 306 nucleotides from the intron 10, preceded by a T repeat of different length (10 and 51 respectively for case 1 and case 2). In both cases, the position of the insertion does not allow defining the specific initial (TG)mTn tract of these alleles. These molecular lesions alter the physiological splicing of the CFTR pre-mRNA. We functionally studied these samples both by RT-PCR, gel electrophoresis and densitometric analysis, as well as by Real-time PCR obtaining, by both approaches, percentages of anomalous transcript over 50%. Considering the very limited contribution of (TG)10T9 allele to exon 10 anomalous splicing (as evidenced both in literature and in our experiments), it appears that these insertions completely suppress the physiological splicing process and cause the loss of functional CFTR protein. Consequently, when paired with a severe CFTR pathogenic variant in trans on the other allele, they originate classic CF phenotypes. These molecular lesions have not been previously described in literature and were found in two patients with similar phenotypes. Their molecular and functional features, allow classifying them as novel CF-causing variants of CFTR. They could be named as c.1210-34TG[10]_1210-34TG[4]ins317 for case 1 and c.1210-34TG[10]_1210-34TG[4]ins358 for case 2. Their selection and functional characterization allowed the completion of the genetic characterization of both patients.

Both pathogenic variants derive from the duplication of the same portion of DNA from intron 10. The poly-T stretch at the beginning of the duplicated portion may had some kind of interaction/recombination with the repeated T of the (TG)mTn tract, which could constitute the first mutational event originating the insertion. Then, a molecular mechanism of divergence from this common ancestral allele can be supposed. Due to the presence of several

repeated T at the beginning of the inserted portion, it is possible that during the meiotic divisions the mutated allele underwent two different rearrangements leading to the formation of two novel pathogenic variants. As a support of this hypothesis, it should be taken into account that both patients have their origin in the same limited geographical area.

Mutational search protocols at high sensitivity and specificity are useful to the structural and functional characterization of CFTR alleles unknown at initial genetic characterization. The discovery of novel rare pathogenic variants of CFTR, as well as their experimental functional characterization, are mandatory to ameliorate our diagnostic, prognostic and, in the era of CF personalized medicine, therapeutic ability.

## Supporting information

**S1 Minimal data set. Densitometric values used for Fig 4A.** .
(XLSX)

**S2 Minimal data set. Confirmatory real time data for Fig 4B.**
(XLSX)

## Acknowledgments

This work was supported by Regione Lazio (research projects 2008–2012).

## Author Contributions

**Conceptualization:** Silvia Pierandrei, Natalia Cirilli, Marco Cipolli, Marco Lucarelli.

**Data curation:** Silvia Pierandrei, Giovanna Blaconà, Benedetta Fabrizzi, Giuseppe Cimino, Natalia Cirilli, Nicole Caporelli, Antonio Angeloni, Marco Cipolli, Marco Lucarelli.

**Formal analysis:** Marco Lucarelli.

**Funding acquisition:** Marco Lucarelli.

**Investigation:** Silvia Pierandrei, Giovanna Blaconà, Benedetta Fabrizzi, Giuseppe Cimino, Natalia Cirilli, Nicole Caporelli, Antonio Angeloni, Marco Cipolli, Marco Lucarelli.

**Methodology:** Silvia Pierandrei, Giovanna Blaconà, Benedetta Fabrizzi, Giuseppe Cimino, Natalia Cirilli, Nicole Caporelli, Antonio Angeloni, Marco Cipolli, Marco Lucarelli.

**Supervision:** Silvia Pierandrei, Marco Lucarelli.

**Validation:** Silvia Pierandrei, Giovanna Blaconà, Marco Lucarelli.

**Writing – original draft:** Silvia Pierandrei, Natalia Cirilli, Marco Lucarelli.

**Writing – review & editing:** Silvia Pierandrei, Marco Lucarelli.

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
