## [Decision Letter · Decision Letter 0]

10 Sep 2019

[EXSCINDED]

Two novel and correlated CF-causing insertions in the (TG)mTn tract of the CFTR gene

PONE-D-19-19571

Dear Dr. Lucarelli,

We are pleased to inform you that your manuscript has been judged scientifically suitable for publication and will be formally accepted for publication once it complies with all outstanding technical requirements.

Based on the reviewers’ comments and my own evaluation, this manuscript meets PLOS ONE’s criteria and can be accepted for publication.

With kind regards,

Emanuele Buratti, Ph.D.

Academic Editor

PLOS ONE

1. We understand that no formal ethical approval was obtained for this research. We also understand that your institutional review board (IRB) may have waived the need to obtain formal approval. Please clarify if this is correct and provide a written statement from your ethics committee as a supplementary file. Please clarify whether all patient data was anonymised prior to analysis by the authors.

Reviewers' comments:

Reviewer's Responses to Questions

**Comments to the Author**

1. Is the manuscript technically sound, and do the data support the conclusions?

Reviewer #1: Yes

Reviewer #2: Yes

2. Has the statistical analysis been performed appropriately and rigorously? 

Reviewer #1: Yes

Reviewer #2: N/A

3. Have the authors made all data underlying the findings in their manuscript fully available?

Reviewer #1: Yes

Reviewer #2: Yes

4. Is the manuscript presented in an intelligible fashion and written in standard English?

Reviewer #1: Yes

Reviewer #2: Yes

5. Review Comments to the Author

Reviewer #1: The manuscript is very clear and reported two novel and related pathogenic variants of the Cystic Fibrosis ransmembrane conductance Regulator (CFTR) gene. Two patients in the same region (South of Marche Region, Central Italy) have alterations that have not been previously described in literature. Both patients with diagnosis of Cystic Fibrosis (CF) are heterozygous p.Phe508del, and in trans the insertion of part of intron 10 in intron 9 of the CFTR gene, within the (TG)m repeat. The molecular lesions resulted to be very similar in both patients, with only a difference in the number of T in the poly-T stretch. After characterization at RNA level, a complete anomalous splicing, without exon 10, from the allele with the insertion of both patients has been shown. Consequently, these alleles with the insertions are expected to contribute to the formation of a non functional CFTR protein.

All figures and their legends are clear and well documented.

For statistical tests, expression data were evaluated using analysis of variance (ANOVA) by GraphPad Prism 5.

A p<0.05 was considered statistically significant. So it's the reason why I think that the statistical analysis been performed appropriately and rigorously.

Reviewer #2: Cystic fibrosis is an important and common genetic disorder especially among Caucasians, and regarding the complexity of phenotypes encountered in patients and cases which remain genetically undiagnosed, it is therefore important to present new mutations which complete our knowledge about the molecular mechanisms of disease occurence and genotype- phenotype correlation.

In the present manuscript, authors presented a new mutation resulting from internal insertion within intron 9 near the TG tract, affecting the processing of RNA. The mutation analysis was well conducted both structurally (at the level of DNA) and functionnally (at the level of cDNA). All sections of the manuscript are well presented.

6. PLOS authors have the option to publish the peer review history of their article (what does this mean?). If published, this will include your full peer review and any attached files.

Reviewer #1: No

Reviewer #2: Yes: Haleh Akhavan-Niaki

---

## [Editor Report · Acceptance letter]

30 Sep 2019

PONE-D-19-19571 

Two novel and correlated CF-causing insertions in the (TG)mTn tract of the CFTR gene 

Dear Dr. Lucarelli:

I am pleased to inform you that your manuscript has been deemed suitable for publication in PLOS ONE. Congratulations! Your manuscript is now with our production department. 

With kind regards,

on behalf of

Dr. Emanuele Buratti 

Academic Editor

PLOS ONE